# Exploring Microbiome Functional Dynamics through Space and Time with Trait-Based Theory

Leonora S. Bittleston,ᵃ Zachary B. Freedman,ᵇ Jessica R. Bernardin,ᵃ Jacob J. Grothjan,ᶜ Erica B. Young,ᶜ Sydne Record,ᵈ Benjamin Baiser,ᵉ Sarah M. Grayᶠ

ᵃDepartment of Biological Sciences, Boise State University, Boise, Idaho, USA

ᵇDepartment of Soil Science, University of Wisconsin-Madison, Madison, Wisconsin, USA

ᶜDepartment of Biological Sciences, University of Wisconsin-Milwaukee, Milwaukee, Wisconsin, USA

ᵈDepartment of Biology, Bryn Mawr College, Bryn Mawr, Pennsylvania, USA

ᵉDepartment of Wildlife Ecology and Conservation, University of Florida, Gainesville, Florida, USA

ᶠUniversity of Fribourg, Department of Biology-Ecology and Evolution, Fribourg, Switzerland

**ABSTRACT** Microbiomes play essential roles in the health and function of animal and plant hosts and drive nutrient cycling across ecosystems. Integrating novel trait-based approaches with ecological theory can facilitate the prediction of microbial functional traits important for ecosystem functioning and health. In particular, the yield-acquisition-stress (Y-A-S) framework considers dominant microbial life history strategies across gradients of resource availability and stress. However, microbiomes are dynamic, and spatial and temporal shifts in taxonomic and trait composition can affect ecosystem functions. We posit that extending the Y-A-S framework to microbiomes during succession and across biogeographic gradients can lead to generalizable rules for how microbiomes and their functions respond to resources and stress across space, time, and diverse ecosystems. We demonstrate the potential of this framework by applying it to the microbiomes hosted by the carnivorous pitcher plant *Sarracenia purpurea*, which have clear successional trajectories and are distributed across a broad climatic gradient.

**KEYWORDS** Y-A-S, biogeography, ecosystem function, microbiome, pitcher plant, succession, trait

## A FUNCTIONAL TRAIT-BASED PERSPECTIVE FOR MICROBIAL ECOLOGY

Understanding general processes controlling the structure and function of microbiomes is a holy grail in microbial ecology. A functional trait-based perspective offers potential to elucidate microbial impacts on ecosystems through space and time to reveal such generalities (1–4). However, a challenge that microbial ecologists face is that trait-based classifications of microorganisms are historically conceptualized from theory generated for plants and animals. For example, the fast-growing, low-yield copiotroph and slow-growing, high-yield oligotroph life histories (5–7) are rooted in the $r$- and $K$-selection of Pianka (8), which categorizes plants and animals into two functional groups based on life span (short for $r$ and long for $K$) and reproductive effort (large for $r$ and small for $K$). Whereas a recent genomic analysis provides support for the copiotroph-oligotroph continuum for microorganisms (9), this division fails to encapsulate the metabolic plasticity of microbes that acquire complex resources or the stress tolerance of microbes adapted to extreme environments, which are prevalent in soils and other systems (10, 11). More recently, several microbial studies (12–14) have adapted Grime's competitor-stress-ruderal (C-S-R) framework, which juxtaposes three life history strategies for plants: competitors (C) capitalize on resource acquisition in productive and undisturbed environments, stress tolerators (S) thrive under sustained stress and low resources, and ruderals (R) exist in recently disturbed but less stressful habitats

Address correspondence to Leonora S. Bittleston, leonorabittleston@boisestate.edu.

Diving into carnivorous pitcher plant microbial communities to explore traits of microbiomes across space and time

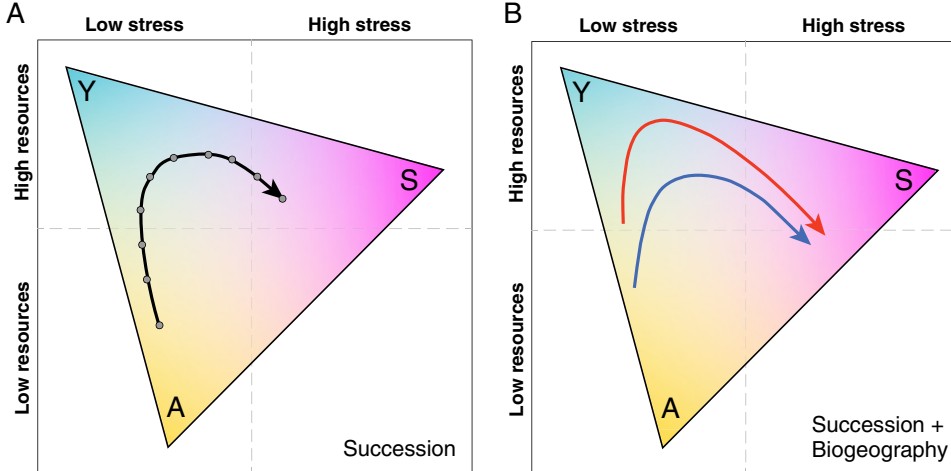

**FIG 1** Hypothetical successional trajectories of microbiomes within the Y-A-S framework. (A) A successional trajectory where each gray dot represents a temporal sample. Low initial resource conditions result in a microbiome composed predominantly of nutrient acquisition (A) traits. Resources accumulate over time, and the microbiome shifts in functional composition toward high-yield (Y) traits before biotic and/or abiotic stressors start to favor stress tolerance (S) traits at the end of the successional trajectory. (B) Successional trajectories for tropical (red line) and temperate (blue line) microbiomes, which under the latitudinal diversity gradient hypothesis would have greater resources in the tropics. The tropical trajectory (red line) starts at a higher resource level and thus with the predominance of (Y) traits, while the temperate system begins in low-resource conditions with a greater representation of (A) traits. The tropical trajectory maintains a greater prevalence of (Y) traits than the temperate trajectory as resources accumulate through time until both trajectories increase in stress tolerator (S) traits toward the end of succession.

(15). However, applying this framework to microbes presents challenges because phototrophic plant traits are not fully comparable to traits for heterotrophic microbes whose reliance on external sources of carbon and energy from the environment drive metabolic and allocation tradeoffs (16). Furthermore, stress responses of microbes to disturbance, which are linked to the amount, availability, and quality of resources, are likely different for heterotrophs compared to plants (16).

The shortcomings of previous functional trait-based frameworks have hindered progress toward a unifying theory of microbiome functional spatiotemporal dynamics. One promising recent framework developed for soil microbiomes advances Grime's C-S-R theory to conceptualize the functional space of microbial assemblages as a triangle where each corner represents a dominant life history strategy, defined by microbiome-centric functional traits (16). One corner represents the microbial high-yield strategy (Y), exhibiting traits central to carbon metabolism, biosynthesis, and high biomass accumulation, with low investment in alternative nutrient acquisition or stress tolerance strategies (16). Another corner represents microbial resource acquisition strategies (A), with defining traits including chemotaxis and the production of extracellular enzymes and siderophores as well as uptake systems and transporters (17). The third corner represents the microbial stress tolerance strategy (S), including traits such as sporulation, damage repair, and maintenance of cellular integrity (Fig. 1). Based on Y-A-S trade-offs, each strategy is successful at different points along the gradients of two important drivers of microbial assembly, resource availability, and stress.

While the Y-A-S framework was proposed to characterize soil microbiome function, using it to confront ecological theory on succession and biogeography holds promise for transforming our understanding of microbiome dynamics across systems. Succession and biogeography of ecological assemblages depend on how organisms respond to temporal and spatial gradients, respectively. We argue that these gradients can be contextualized within two axes, stress and resources, which are linked to the function of microbial assemblages through the Y-A-S trait-based life history strategies. Thus, if we understand how gradients of stress and resources change through time and space, we should be able to predict the successional trajectory and biogeographic relationships of microbiome function.

To illustrate the temporal aspect (Fig. 1A), a hypothetical successional trajectory begins with a microbiome composed predominantly of nutrient acquisition (A) traits in response to low initial resource conditions. Over time, resources accumulate and the microbiome shifts in functional composition toward high-yield (Y) traits before factors such as predation and anaerobic, oxidative, or climatic stress start to favor stress tolerance (S) traits at the end of the successional trajectory. Microbiomes can take many successional trajectories through the Y-A-S triangle depending on how resource availability and stress change through time in the system. For example, microbiomes in ecosystems with stressful conditions (e.g., deserts, hot springs) may have successional trajectories that begin with a greater proportion of stress tolerator traits (S) (18, 19). Alternatively, ecosystems in which resources decline drastically in the later stages of succession will likely have increased competition and show a predominance of nutrient acquisition (A) traits toward the end of the successional trajectory.

## TRAIT-BASED APPROACHES WITH MACROECOLOGICAL THEORY

Further extending the Y-A-S framework across a biogeographical gradient, we consider the latitudinal diversity gradient (LDG), where decreases in species richness from the equator to the poles have been well-established for many plants and animals (20, 21), although there is mixed support for microbes (22, 23). Hypotheses for the LDG (e.g., greater productivity, more stable climate, warmer temperatures in the tropics, etc.) can be reframed along axes of stress and resource availability to enable predictions for how microbiome functional succession can change with latitude in the Y-A-S framework (24). Using the hypothetical successional trajectory from Fig. 1A and the LDG hypothesis of greater resources (i.e., productivity) in the tropics (20, 25, 26), the tropical trajectory (Fig. 1B, red line) starts at a higher resource level and thus consists primarily of (Y) life history traits, whereas the temperate system (Fig. 1B, blue line) begins in low-resource conditions with greater predominance of (A) traits. Both trajectories have an increase in the proportion of (Y) life history traits as resources accumulate through time, but the tropical trajectory maintains a greater prevalence of (Y) traits than the temperate trajectory due to greater resource availability in the tropics. Finally, both trajectories increase in stress tolerator (S) traits toward the end of succession as competition and higher cell density lead to abiotic and biotic stress (Fig. 1B). This stress may include reduced supply of readily usable organic carbon sources, build-up of toxic by-products, pH changes, and declining oxygen concentrations (27, 28), as well as increased viral load and grazing pressure by invertebrates (29).

## EFFECTIVE APPLICATION OF THE Y-A-S FRAMEWORK

The Y-A-S framework will be most useful when researchers can accurately quantify microbial population and community traits across systems. This poses significant challenges, as methodological differences and informative traits for each life-history strategy will likely vary depending on the system. For example, the current metrics for yield-based traits, like microbial growth efficiency (MGE) and carbon use efficiency (CUE), can be quantified using several methods that differ in technical variability and make cross-study and cross-system comparison more difficult (30). In addition to physiological measures for traits (i.e., MGE and enzyme activity), genomic data will help to advance how we estimate and model community function. For example, a shotgun metagenomic analysis makes it possible to estimate the functional potential of a whole community, while metatranscriptomic data can provide a snapshot of gene expression reflecting current function. While recent evidence supports quantifying yield- and stress-based traits by genomic metabolic markers (i.e., osmoregulation, sporulation, cell repair) and genome size (31–33), significant challenges remain for using metagenomic and transcriptomic data due to incomplete functional databases and gene annotations.

The existing, abundant amplicon (e.g., 16S rRNA) sequencing data can be used as a proxy for functional potential (34), although additional difficulties arise. For example, 16S rRNA copy number could help predict life strategies and growth rates of bacterial

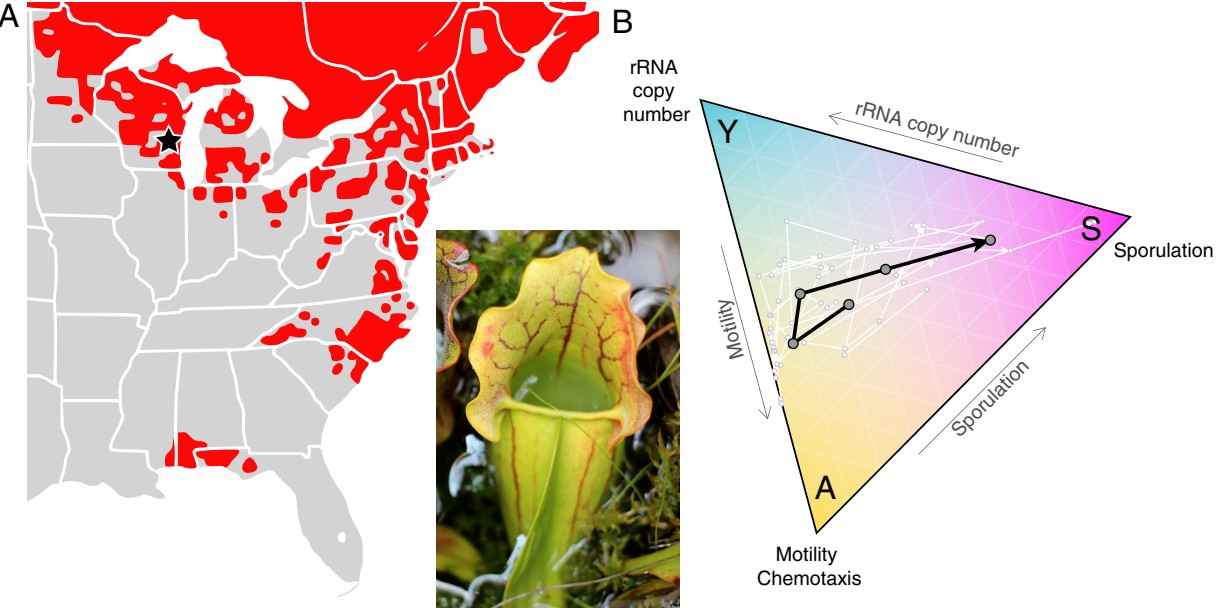

FIG 2 (A) Distribution of *S. purpurea* in red (Noah Elhardt, public domain) with a pitcher (inset; photograph by L. Bittleston). (B) An example of how *S. purpurea* pitcher microbiomes move through the Y-A-S functional space during succession. This ternary plot uses PICRUSt2 functional output of 16S rRNA data averaged over 10 pitchers in a single wetland population in Wisconsin (black star in panel A). Dots are samples from 3, 7, 14, 28, and 62 days after pitcher-opening, and arrows indicate the direction of time. The white lines connect the data from each pitcher, while the black line shows the mean across all 10 pitchers. For this preliminary test, we chose single traits to represent each strategy. For motility, we summed the normalized abundances of all genes at the KO level that contained "chemotaxis protein" in their descriptions. Similarly, for sporulation, we summed all that contained "sporulation protein." For a community measure of rRNA copy number, we weighted the estimated rRNA copy number for each amplicon sequence variant (ASV) by its relative abundance. We recognize that rRNA copy number is not an ideal trait to represent the Y strategy, as it is correlated with maximum growth rate, which does not always translate to higher yield (4, 16); furthermore, it is often not accurately predicted by 16S data (48). A measure of microbial growth efficiency would be a better Y trait; however, it was not available for these data. Data and code are available at https://doi.org/10.7910/DVN/Z0FQK7.

taxa (4, 35, 36). Furthermore, bacterial composition paired with known taxon functions have been used to extrapolate to human microbiome functions (37). However, 16S rRNA estimations are often poorly correlated with functional profiles from shotgun metagenomes, particularly outside human data sets (38). The paucity of functional information for environmental taxa means that more direct functional assessments using metagenome and metatranscriptome approaches will likely provide more detailed community profiling (39) to enable cross-system trait-based inquiry to microbiome functional dynamics over space and time.

Analyzing metagenomic and metatranscriptomic data within the Y-A-S framework will allow ecologists to improve functional trait databases and understand microbial functional patterns that span large spatial and temporal gradients (36, 40). Furthermore, continued collaboration between microbial ecologists and macrobiologists (e.g., through the National Science Foundation's emerging Center for Advancement and Synthesis of Open Environmental Data and Sciences) will help define the key ecological processes driving microbiome assembly.

### BOX 1: *SARRACENIA PURPUREA* MICROBIOME CASE STUDY

The *Sarracenia purpurea* microbiome is an ideal model system for incorporating succession and biogeography into the Y-A-S framework (Fig. 2). This plant species grows in nutrient-poor habitats and relies on the microbiomes that inhabit its water-filled, pitcher-shaped leaves to transform captured insect prey into usable mineral nutrients. The food web is composed of bacteria, protozoa, rotifers, and dipteran larvae, and the microbiome is thought to provide essential "ecosystem services" for the host plant via degradation of prey (29, 41–44). The leaves of the plant are sterile

when they open, and the recruitment of their microbiome represents the start of succession, which can be followed throughout the growing season of the plant (45, 46). *S. purpurea* inhabits a large latitudinal gradient, ranging from Florida to Canada (23) (Fig. 2A), with microbiome richness exhibiting a negative relationship with latitude that is predicted by the latitudinal diversity gradient (LDG) (23). The wide distribution and observed LDG provide opportunities to examine biogeographical factors driving microbiome functions. Early in microbiome succession, we can predict dominance of acquisition strategies (A) because resources are scarce in newly opened leaves. The trajectory will then depend on the resources and stress encountered, which may vary with latitude. Microbial composition and function could change at a higher rate in warmer climates, related to microbial metabolism and increased predation rates from higher trophic levels.

As a test case to examine successional trajectories within the Y-A-S framework, we followed microbial communities over 2 months in 10 individual *S. purpurea* pitchers from the Cedarburg Bog (47). On days 3, 7, 14, 28, and 62 after the leaves had opened into their characteristic pitcher shape, 3 ml of natural pitcher water was collected from each of the 10 pitchers using a sterile syringe and tubing (43), and community DNA was extracted for sequencing (47). 16S rRNA V3_4 sequence data were processed with QIIME2 version 2020.2 and used for PICRUSt2 analysis (34) to generate predictions of microbiome functions. Changes in relative abundances of estimates for three functional traits, rRNA community copy number as Y, motility and chemotaxis proteins as A, and sporulation proteins as S, were mapped over time (Fig. 2B; data and code are available via the Harvard Dataverse at https://doi.org/10.7910/DVN/Z0FQK7). Successional trajectories varied across the 10 pitchers (Fig. 2B) but on average began near the center of the triangle, but closest to A. Nutrient acquisition strategies then became less dominant, and the microbiomes moved toward Y, potentially due to resources (i.e., insect prey) entering the system. Later, the trajectory moved toward S, as competition likely increased while readily available resources decreased (Fig. 2B). To our knowledge, this is the first application of the Y-A-S framework to empirical data beyond soil systems.

With future application of more complex and accurate metagenomic and metatranscriptomic data from microbiomes during succession and across a latitudinal gradient, the Y-A-S framework will facilitate examination of functional changes in microbiomes across temporal and spatial scales. A multivariate approach across numerous traits will allow us to identify the most important suites of functional traits that are active during succession and define life history strategies across broad biogeographical scales.

## ACKNOWLEDGMENTS

We thank the editors and reviewers for their valuable feedback.

We acknowledge funding from the National Science Foundation (NSF) through an Understanding the Rules of Life-Microbiome Theory and Mechanisms 2 collaborative grant to L.S.B., S.M.G., Z.B.F., E.B.Y., and B.B. (awards 2025250, 2025262, 2025337, and 2025110). In addition, L.S.B. was funded by the NSF Idaho EPSCoR Program and by the NSF under award number OIA-1757324; S.R. was supported by the Bryn Mawr K.G. Research Fund; E.B.Y. was funded by a UWM-DIG grant; the establishment of the *Sarracenia purpurea* International Network (SPIN) was funded by the Swiss National Science Foundation (SNSF, grant IZSEZ0_186214) to S.M.G.

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
