## [Reviewer comments · mSystems]

Exploring microbiome functional dynamics through space and time with trait-based theory

Leonora Bittleston, Zachary Freedman, Jessica Bernardin, Jacob Grothjan, Erica Young, Sydne Record, Benjamin Baiser, and Sarah Gray

Corresponding Author(s): Leonora Bittleston, Boise State University

Review Timeline:

Submission Date:	April 29, 2021
Editorial Decision:	May 24, 2021
Revision Received:	June 30, 2021
Accepted:	July 12, 2021

Editor: Ashley Shade

Reviewer(s): Disclosure of reviewer identity is with reference to reviewer comments included in decision letter(s). The following individuals involved in review of your submission have agreed to reveal their identity: Ashish Malik (Reviewer #2)

Transaction Report:

DOI: <https://doi.org/10.1128/mSystems.00530-21>

May 24, 2021

Dr. Leonora S Bittleston
Boise State University
Biological Sciences
1910 University Drive
Boise, ID 83725

Re: mSystems00530-21 (Exploring microbiome functional dynamics through space and time with trait-based theory)

Dear Dr. Leonora S Bittleston:

Thank you for submitting your manuscript to mSystems. We have completed our review and I am pleased to inform you that, in principle, we expect to accept it for publication in mSystems. However, acceptance will not be final until you have adequately addressed the reviewer comments.

Thank you for the privilege of reviewing your work. Below you will find instructions from the mSystemseitorial office and comments generated during the review.

Preparing Revision Guidelines

For complete guidelines on revision requirements, please see the Instructions to Authors at <https://msystems.asm.org/sites/default/files/additional-assets/mSys-ITA.pdf>. **Submissions of a paper that does not conform to mSystems guidelines will delay acceptance of your manuscript.**

Sincerely,

Ashley Shade

Editor, mSystems

Journals Department
Reviewer comments:

Reviewer #1 (Comments for the Author):

Bittleston and colleagues present one of the first tests of the Yield-Acquisition-Stress (YAS) hypothesis of microbiome-specific trait tradeoffs. In their study system, bacteria within pitcher plants first exhibit acquisition traits, then yield traits and finally stress traits. This paper is an elegant proof of concept but could be improved for clarity in several sections.

(1) Lines 79-86. This seems like a likely pathway of succession but not the only one. Why not exhibit stress-like traits in the beginning of succession if the environmental is stressful - aka hot springs/desert? Why not exhibit competitive traits near the end? Analogs to succession of other ecological communities may help here.

(2) Lines 87-103. Here the YAS framework seems like a bit of an over-simplification. It seems like the tropical and temperate communities should start in different parts of the YAS triangle based off of resource limitation (tropical soils) or stressful conditions (boreal soils) and then cycle through the phases described here. I also suggest that you cite Dickey et al. 2021 *Frontiers in Ecology and Evolution* for a comprehensive review of LDG in microbiomes.

(3) Lines 148-162. This test of the YAS framework is an interesting approach, but should be more fleshed out in the main text. Many of the study details are hidden in the figure 2 legend like study design and analysis. If ten samples were averaged, it seems like there should be either 10 lines or error bars around the average line. Some detail about sequencing and bioinformatic should also be included, ideally as a supplement.

Minor concerns.

Figure 2. I believe the red outlines are the pitcher plant range but it's not completely clear. Please add this to the legend.

Reviewer #2 (Comments for the Author):

This perspective piece provides a conceptual extension of the YAS trait-based framework to assess how microbiome functions change across temporal and spatial scales. Authors argue that "if we understand how gradients of stress and resources change through time and space, we should be able to predict the successional trajectory and biogeographic relationships of microbiome function" which is an attractive testable theory.

Overall, the short article makes a good case for investigating microbiome function across space and time using generalised ecological theory on the basis of the YAS framework. Authors provide a theoretical basis to the concepts of YAS trait changes during succession and across biogeography, and then provide a case study to demonstrate YAS trajectory during microbiome succession.

The case study presented is a great example to demonstrate trait changes over time. I am curious how the generated data was plotted in the three-dimensional trait space. I understand this is not a data paper but there has to be a basis on which the case study data is presented. I don't see any citations of previously published data paper that was used to generate the figure and the concepts.

Authors mention that there could be biogeographic patterns to such trait distributions but provide no specific predictions on the basis of the generalised theory presented earlier in the article.

Other points

I wonder if having section headings will improve the flow and readability of the paper.

line 24: functional traits, line 28: trait composition can affect functions. Need consistency in usage. As far as I understand, trait is an index of functions.

Line 90: long windy sentence, consider revising

Line 93: Authors refer to "microbiome functional succession changes across latitude". After reading the rest of the paragraph it appears that the authors mean that successional patterns differ depending on the latitude. Fig. 1B is labelled as biogeography which also creates a confusion, because it actually is succession across space. Consider revising to provide clarity.

In line 101 and earlier, it is not clear how competition and higher cell density can lead to abiotic stress. In ecological succession, it can be easy to visualise systems moving from too little to optimum to too much. Is there scope to discuss such changes in stress levels due to abiotic factors through an example?

Paragraph starting line 104 and 119 are just technical aspects of measuring YAS traits and aren't really making any novel contributions.

Dear *mSystems* Editorial Team,

We thank the editor and reviewers for constructive comments to improve our manuscript entitled, “*Exploring microbiome functional dynamics through space and time with trait-based theory*”. We have done our best to address all of the reviewers’ comments, which we outline in our point-by-point response below. In particular, we provided greater detail on the methodology underlying the pitcher plant example, revised both figures and included the data and code for Figure 2B, and clarified our hypothetical Y-A-S successional trajectories outlined in Figure 1.

Please feel free to reach out to us with any questions or additional comments.

Sincerely,
Leonora Bittleston (on behalf of my co-authors)

Below the reviewer comments are shown in black text and our point by point responses to their comments are shown in blue. Line numbers in our responses correspond to the final version of the manuscript without track changes.

Reviewer comments:

Reviewer #1 (Comments for the Author):

Bittleston and colleagues present one of the first tests of the Yield-Acquisition-Stress (YAS) hypothesis of microbiome-specific trait tradeoffs. In their study system, bacteria within pitcher plants first exhibit acquisition traits, then yield traits and finally stress traits. This paper is an elegant proof of concept but could be improved for clarity in several sections.

(1) Lines 79-86. This seems like a likely pathway of succession but not the only one. Why not exhibit stress-like traits in the beginning of succession if the environmental is stressful - aka hot springs/desert? Why not exhibit competitive traits near the end? Analogs to succession of other ecological communities may help here.

We appreciate this and other constructive comments related to our perspective. We acknowledge that the hypothetical trajectory presented here and in Fig. 1a is only one potential trajectory and that there may be other successional trajectories that a community may follow. To emphasize this, we added the following text to this section (lines 88-92):

“For example, microbiomes in ecosystems with stressful conditions (e.g., deserts, hot springs) may have successional trajectories that begin with a greater proportion of stress tolerator traits (S) (18, 19). Alternatively, ecosystems in which resources decline drastically in the later stages of succession will likely have increased competition and show a predominance of nutrient acquisition (A) traits towards the end of the successional trajectory.”

(2) Lines 87-103. Here the YAS framework seems like a bit of an over-simplification. It seems like the tropical and temperate communities should start in different parts of the YAS triangle based off of resource limitation (tropical soils) or stressful conditions (boreal soils) and then cycle through the phases described here. I also suggest that you cite Dickey et al. 2021 *Frontiers in Ecology and Evolution* for a comprehensive review of LDG in microbiomes.

We thank the reviewer for their thoughtful comment and for highlighting the highly relevant Dickey et al., 2021 citation, which we have now cited in the text. We also agree that Y-A-S can seem like a slight oversimplification here and that other possible hypothetical successional trajectories can differ from those described here. For clarity, in the revised manuscript we emphasize that the LDG example is hypothetical, that support for the LDG is mixed for microbiomes, and that other hypothetical successional paths through Y-A-S are possible. We also revised the label in Fig. 1b to clarify that the Y-A-S triangle featured in it represents both succession and biogeography to emphasize to the readers that Fig. 1b is an extension of the successional trajectory in Fig. 1a, which we have added text about caveats to per Reviewer 1’s first comment. In Box 1 we also emphasize that the *S. purpurea* microbiome richness exhibits a negative relationship with latitude, in support of the LDG. The revised text now reads:

Lines 94-97:

“Further extending the Y-A-S framework across a biogeographical gradient, we hypothetically consider the Latitudinal Diversity Gradient (LDG) where decreases in species richness from the equator to the poles have been well-established for many plants and animals (20, 21), although there is mixed support for microbes (22, 23).”

Lines 154-158:

“*S. purpurea* inhabits a large latitudinal gradient, ranging from Florida to Canada (23); Fig. 2A) with microbiome richness exhibiting a negative relationship with latitude that is predicted by the latitudinal diversity gradient (LDG) (23). The wide distribution and observed LDG provides opportunities to examine biogeographical factors driving microbiome functions.”

(3) Lines 148-162. This test of the YAS framework is an interesting approach, but should be more fleshed out in the main text. Many of the study details are hidden in the figure 2 legend like study design and analysis. If ten samples were averaged, it seems like there should be either 10 lines or error bars around the average line. Some detail about sequencing and bioinformatic should also be included, ideally as a supplement.

We have updated the text in this section to explain sampling context, experimental details, and bioinformatics (lines 164-178):

“As a test case to examine successional trajectories within the Y-A-S framework, we followed microbial communities over two months in ten individual *S. purpurea* pitchers from the Cedarburg Bog (47). On days 3, 7, 14, 28 and 62 after the leaves had opened into their characteristic pitcher shape, 3 mL of natural pitcher water was collected from each of the ten pitchers using a sterile syringe and tubing (43) and community DNA was extracted for sequencing (47). 16S rRNA V3_4 sequence data were processed through QIIME2 Version 2020.2 and used for PICRUST2 analysis (34) to generate predictions of microbiome functions. Changes in relative abundances of estimates for three functional traits: rRNA community copy number as Y, motility and chemotaxis proteins as A, and sporulation proteins as S, were mapped over time (Fig. 2B; data and code are available via the Harvard Dataverse: <https://doi.org/10.7910/DVN/Z0FQK7>). Successional trajectories varied across the 10 pitchers (Fig. 2B) but on average began near the center of the triangle, but closest to A. Nutrient acquisition strategies then became less dominant, and the microbiomes moved towards Y; potentially due to resources (i.e., insect prey) entering the system. Later, the trajectory moved towards S, as competition likely increased while readily available resources decreased (Fig. 2B).”

We also uploaded the data and code used to generate the ternary plot shown in Fig. 2B and have included the link to the Harvard Dataverse Repository. The available files include the R code, the Amplicon Sequence Variant (ASV) table, the relevant metadata, and two files from PICRUST2: the normalized, predicted KEGG orthologs (KO) file and the estimated ribosomal RNA copy numbers.

We agree that showing the variation in the successional trajectories across the 10 pitchers is valuable and we have updated Figure 2B and its legend to include both plotted lines for each of the 10 pitchers (in white) as well as the mean values (in black), which we hope more clearly portrays both the variability and the averaged trend. We have chosen to keep the case study as a box, as this format was recommended by the editors.

Also, we note that the addition of this information to the main text, along with other edits to address comments, brings the text of the paper 374 words over the 1500 word limit.

For now, we have left this added text in the main text as both reviewers requested more information on this proof-of-concept analysis. However, we defer to the editor in terms of whether to put some of this text into a supplement.

Minor concerns.

Figure 2. I believe the red outlines are the pitcher plant range but it's not completely clear. Please add this to the legend.

Thank you for pointing out this issue. We have changed the text for the Figure 2A legend to now read: "Distribution of *S. purpurea* in red".

Reviewer #2 (Comments for the Author):

This perspective piece provides a conceptual extension of the YAS trait-based framework to assess how microbiome functions change across temporal and spatial scales. Authors argue that "if we understand how gradients of stress and resources change through time and space, we should be able to predict the successional trajectory and biogeographic relationships of microbiome function" which is attractive testable theory.

Overall, the short article makes a good case for investigating microbiome function across space and time using generalised ecological theory on the basis of the YAS framework. Authors provide a theoretical basis to the concepts of YAS trait changes during succession and across biogeography, and then provide a case study to demonstrate YAS trajectory during microbiome succession.

The case study presented is a great example to demonstrate trait changes over time. I am curious how the generated data was plotted in the three-dimensional trait space. I understand this is not a data paper but there has to be a basis on which the case study data is presented. I don't see any citations of previously published data paper that was used to generate the figure and the concepts.

Thank you for the suggestion, we have now added text to explain the sampling context, experimental details and bioinformatics (lines 164-178). We also uploaded the data and code used to generate the ternary plot shown in Fig. 2 and have included the DOI and link to the Harvard Dataverse repository.

Authors mention that there could be biogeographic patterns to such trait distributions but provide no specific predictions on the basis of the generalised theory presented earlier in the article.

In the revised manuscript, we clarified and emphasized our theoretical LDG example (Box 1) and included a citation to a recently published manuscript for readers that may be interested in further reading about this topic (Dickey et al., 2021). In Box 1 we also emphasize that the *S. purpurea* microbiome richness exhibits a negative relationship with latitude, in support of the LDG, and emphasize predictions that can be made from generalized theory earlier in the manuscript. The revised text now reads:

Lines 94-97:

“Further extending the Y-A-S framework across a biogeographical gradient, we hypothetically consider the Latitudinal Diversity Gradient (LDG) where decreases in species richness from the equator to the poles have been well-established for many plants and animals (20, 21), although there is mixed support for microbes (22, 23).”

Lines 88-92:

“For example, microbiomes in ecosystems with stressful conditions (e.g., deserts, hot springs) may have successional trajectories that begin with a greater proportion of stress tolerator traits (S) (18, 19). Alternatively, ecosystems in which resources decline drastically in the later stages of succession will likely have increased competition and show a predominance of nutrient acquisition (A) traits towards the end of the successional trajectory.”

Lines 154-158:

“*S. purpurea* inhabits a large latitudinal gradient, ranging from Florida to Canada (23); Fig. 2A) with microbiome richness exhibiting a negative relationship with latitude that is predicted by the latitudinal diversity gradient (LDG) (23). The wide distribution and observed LDG provides opportunities to examine biogeographical factors driving microbiome functions.”

Lines 158-163:

“Early in microbiome succession, we can predict dominance of acquisition strategies (A) because resources are scarce in newly opened leaves. The trajectory will then depend on the resources and stress encountered, which may vary with latitude. Microbial composition and function could change at a faster rate in warmer climates related to microbial metabolism and increased predation rates from higher trophic levels.”

Other points

I wonder if having section headings will improve the flow and readability of the paper.

We have added headings to corresponding sections to increase clarity and readability.

line 24: functional traits, line 28: trait composition can affect functions. Need consistency in usage. As far as I understand, trait is an index of functions.

Thank you for noticing this issue, we have updated the text on lines 28-29 to read: “However, microbiomes are dynamic, and spatial and temporal shifts in taxonomic and trait composition can affect **ecosystem** functions” (bolded here to highlight the change).

Line 90: long windy sentence, consider revising

We have revised the sentence. It now reads as, “Hypotheses for the LDG (e.g., greater productivity, more stable climate, warmer temperatures in the tropics, etc.) can be reframed along axes of stress and resource availability to enable predictions for how microbiome functional succession can change with latitude in the Y-A-S framework (24).” Lines 97-100.

Line 93: Authors refer to "microbiome functional succession changes across latitude". After reading the rest of the paragraph it appears that the authors mean that successional patterns differ depending on the latitude. Fig. 1B is labelled as biogeography which also creates a confusion, because it actually is succession across space. Consider revising to provide clarity.

We thank the reviewer for this point. We have revised Figure 1B to say “Succession + Biogeography” to clarify that it is a combination of different processes. We feel that this Y-A-S trait-based approach can apply to understanding microbiomes across broader biogeographical comparisons, but the example we provide and have data to support is specific to latitude (Freedman et al. 2021). Hence, we have included biogeography in the more general parts of the manuscript but specifically describe latitude in our more detailed discussion.

In line 101 and earlier, it is not clear how competition and higher cell density can lead to abiotic stress. In ecological succession, it can be easy to visualise systems moving from too little to optimum to too much. Is there scope to discuss such changes in stress levels due to abiotic factors through an example?

We thank the reviewer for the comment so we can clarify this point with some examples including how increased cell density in microbial communities can more generally lead to stress in relation to increased competition for resources and for potential build-up of waste products. Within the *S. purpurea* system, this has been specifically documented as reduced oxygen supply in late succession, high density communities.

We have added some text to clarify this point (Lines 109-112) and include both a general review reference and an example specific to *S. purpurea* microbiome communities:

“This stress may include reduced supply of readily-usable organic carbon sources, build-up of toxic by-products, pH changes, declining oxygen concentrations (27, 28) as well as increased viral load and grazing pressure by invertebrates (29).”

Paragraph starting line 104 and 119 are just technical aspects of measuring YAS traits and aren't really making any novel contributions.

The editors of this special issue noted that this might be many readers' first introduction to the Y-A-S framework, and so we chose to go over how it can be used and potential issues that may arise. We also included some important caveats to the choices of traits, and feel that these paragraphs could be helpful to people who want to apply the framework.

July 12, 2021

Dr. Leonora S Bittleston
Boise State University
Biological Sciences
1910 University Drive
Boise, ID 83725

Re: mSystems00530-21R1 (Exploring microbiome functional dynamics through space and time with trait-based theory)

Dear Dr. Leonora S Bittleston:

Your manuscript has been accepted, and I am forwarding it to the ASM Journals Department for publication. For your reference, ASM Journals' address is given below. Before it can be scheduled for publication, your manuscript will be checked by the mSystems senior production editor, Ellie Ghatineh, to make sure that all elements meet the technical requirements for publication. She will contact you if anything needs to be revised before copyediting and production can begin. Otherwise, you will be notified when your proofs are ready to be viewed.

As an open-access publication, mSystems receives no financial support from paid subscriptions and depends on authors' prompt payment of publication fees as soon as their articles are accepted. =

Publication Fees:

We recognize that the video files can become quite large, and so to avoid quality loss ASM suggests sending the video file via <https://www.wetransfer.com/>. When you have a final version of the video and the still ready to share, please send it to Ellie Ghatineh at eghatineh@asmusa.org.

Sincerely,

Ashley Shade
Editor, mSystems

Journals Department
Phone: 1-202-942-9338